# A Proposed Global Medicines Agency (GMA) to Make Biological Drugs Accessible: Starting with the League of Arab States

**DOI:** 10.3390/healthcare11142075

**Published:** 2023-07-20

**Authors:** Sarfaraz K. Niazi

**Affiliations:** College of Pharmacy, University of Illinois, Chicago, IL 60612, USA; niazi@niazi.com

**Keywords:** biosimilars, Arab states, regulatory guideline, harmonization, centralized approval, EMA, FDA, Global Medicine Authority (GMA)

## Abstract

Medical anthropology teaches us of historical disparity in the accessibility of medicines in the developing world due to their lack of availability and affordability, more particularly of biological drugs, including therapeutic proteins, gene therapy, CRISPR-Cas9, mRNA therapeutics, CART therapy, and many more. This challenge can be resolved by establishing an independent regulatory agency, proposed as the Global Medicines Agency (GMA), with a charter to allow originators from the Stringent Regulatory Agency (SRA) countries to receive immediate registrations applicable to all member states, expanding the market potential as an incentive. For non-SRA countries, it will be limited to biological drugs that are allowed their copies to be made, only biosimilars. A transparent approval process will involve using a rapporteur, a third-party product-related current Good Manufacturing Practice (cGMP), and assurance of the integrity of samples tested for analytical similarity and clinical pharmacology testing. GMA membership will be open to all countries. Still, it is suggested that the League of Arab States, representing 22 states with a population of 400 million, takes the lead due to their cultural and language homogeneity, which is likely to provide a concurrence among the member states. However, some states, like the Gulf Cooperative Council, are already accustomed to this approach, albeit with a different perspective. The target drugs are biotechnology and gene therapy pharmaceuticals, and their scope can be expanded to any drug.

## 1. Introduction

Medical anthropology teaches that health and medicine’s social, cultural, and political dimensions do not favor developing countries, and this disparity grows as newer high-cost drugs enter the market, particularly biotechnological drugs or biologics that are derived from living organisms or containing components of living organisms. Biologics can be composed of sugars, proteins, nucleic acids, or complex combinations of these substances, or may be living entities such as cells and tissues. Biologics are isolated from various natural sources, such as human, animal, or microorganism, and may be produced by biotechnological methods and other cutting-edge technologies [1]. Gene-based and cellular biologics, for example, are often at the forefront of biomedical research and may be used to treat untreatable diseases. Biologics are complex molecules that are not easily characterized; examples include vaccines that are centuries old such as powdered scab inoculations against smallpox that were used in China as early as the 10th century [2]. As these products expanded, a new terminology became necessary to differentiate them from chemical drugs in the early 20th century. This led to a rush to standardize their definition, production, and quality, ultimately resulting in the Biologics Control Act, which the US Congress passed in 1902 [3]. Soon after, biological drugs were expanded to include products made by a biological process or containing a biological entity. This led to therapeutic proteins and genomic medicines, but their production controls have resulted in an exponential increase in development and production costs.

Genomic medicine has dramatically matured in terms of its technical capabilities; however, its accessibility worldwide faces significant barriers beyond mere access to technology. Development strategies for global genomic medicine should recognize an individual country’s pressing public health priorities and disease burdens. Therefore, it is more pragmatic to transfer technology at different stages, from the bottom up, instead of the top down. An à la carte model of global innovation and development strategy offers multiple entry points into the global genomics innovation ecosystem for developing countries, regardless of whether extensive and expensive discovery infrastructure is already in place. [4,5]

More than half of the projected increase in global spending on medicines, which is expected to reach approximately USD 2 trillion in 2027 from an estimated USD 1.5 trillion today [6,7], is attributed to biological drugs such as vaccines, blood, blood components, allergenic, somatic cells, gene therapy, tissues, and recombinant therapeutic proteins.

As the number and variety of biologics increased throughout the 20th century, so did our ability to produce them. In 1949, researchers at Boston Children’s Hospital successfully used an in vitro system to produce Lansing Type II poliovirus using a human tissue cell culture. This development pave the way for the creation of modern biologics. However, the emergence of genetic engineering in the late 1970s and early 1980s created new opportunities for the research and manufacturing of biologics. Researchers could alter existing agents’ DNA sequences to increase their stability, safety, and effectiveness. These modifications may also affect the targeting specificity of the agent, broadening the applicability of some agent types, including antibodies. Finally, genetic engineering has provided scientists with a broader range of feasible production models. Before the development of transfection and transduction, the ability to create a cellular factory was constrained by the production cell’s genome or, in the case of viral production, by the virion’s susceptibility to persistent but non-lethal infection. However, theoretically, any cell could now be made to produce any molecular or protein-based agent. CRISPR technology, for instance, can modify recombinant cell lines to create proteins with specific properties [8].

Since the 1980s, there has been a significant increase in biologics research and production, leading to notable advancements in the creation of novel therapeutic strategies for diseases, including cancer, immunological disorders, and rare genetic illnesses, among others [6]. The creation, manufacturing, and synthesis of a wide variety of complex designer molecular, protein, gene, cell, and tissue-based agents capable of highly selective targeting have expanded the scope of science, which was once limited to the extraction of naturally occurring chemicals [9].

New drug development costs now run into billions of dollars [10], leading to their high prices for at least 12 years for new biological drugs during the exclusivity period [8]. The increased development cost comes from more stringent SRA guidelines necessitated by knowledge about their potency and side effects [11], resulting in the high price of these products (Table 1) to amortize the investment cost. These drugs should be the first focus for securing their entry into developing countries. This applies to both types of drugs, one that can be copied as biosimilars and other biologics, such as vaccines and gene therapy, for which there is no guideline to produce copies. The latter class is more relevant for manufacturing these products in non-SRA countries. While many recent gene therapy products can cost millions of dollars, therapeutic proteins, a type of drug allowed to have biosimilars, may still cost hundreds of thousands of USD. The latter category is, therefore, most suitable for domestic manufacturing. Notably, the GMA will not approve a new drug of any type that has not been approved in an SRA country.

## 2. The GMA Charter

While new drugs have significantly helped affluent countries, the accessibility of these products remains limited in most developing countries for several reasons:Access and affordability: Individuals and communities face challenges in accessing biotechnology drugs in resource-limited settings. This includes analyzing factors such as cost, availability, distribution, and the role of pharmaceutical companies and government policies;Cultural and social dimensions: How biotechnology drugs are perceived, understood, and utilized within specific cultural contexts. This involves exploring local beliefs, practices, and values related to health, illness, and treatment options, as well as the influence of social and cultural factors on the acceptance and use of these drugs;Global health disparities: Analyzing the implications of global health disparities in the distribution and availability of biotechnology drugs. They investigate the power dynamics between developed and developing countries, pharmaceutical companies, and regulatory bodies and their impact on access to these medications.Ethical considerations: Examining the ethical dimensions of developing, testing, and distributing biotechnology drugs in developing countries. This includes exploring issues of informed consent, clinical trials, and the involvement of vulnerable populations;Domestic manufacturing: The cost of establishing the manufacturing of biotechnology drugs keeps developing countries from becoming self-sufficient in their supply. However, this option applies only to those drugs that can be copied in the SRA countries; an example is biosimilars.

The proposed plan to establish a GMA has the following objectives:To make these drugs available in developing countries by simplifying the registration process of novel SRA country products;To encourage SRA originators and domestic developers by making a large market population with a single registration;To ensure data integrity by not requiring all countries to acquire registration dossiers.To ensure the safety and efficacy of products manufactured in non-SRA countries by adopting a rational and practical registration process.

However, the idea of the GMA is not new, as several agencies have been formed with similar goals in the past. Still, the disparity in accessibility of biotechnology drugs and many chemical drugs remains in place. The charter of the GMA is based on learning from the working of these agencies to enable reaching the goal by bringing clarity, rationality, and practicality to the registration of drugs to overcome their accessibility issues.

### 2.1. Regulatory Agencies

The European Medicines Agency (EMA) is responsible for the scientific evaluation, supervision, and safety monitoring of medicines, particularly biotechnology drugs and medicines. However, countries within the EU/EEA can also have their own national regulatory authorities that evaluate and approve medicines for their specific jurisdictions. These national regulatory authorities work with the EMA and may require additional steps for local registration, such as translation, labeling adaptations, or specific national requirements. Some medicines, especially those under national procedures or certain categories, may undergo decentralized or mutual recognition procedures, where individual countries review and grant marketing authorizations based on common assessments and agreements. [https://www.ema.europa.eu/ (accessed on 15 July 2023)].

Pan American Health Organization (PAHO)—PAHO serves as the regional office for the Americas and collaborates with regulatory authorities across the region to facilitate the approval and regulation of medical products; it serves as the regional office for the Americas under the umbrella of the World Health Organization (WHO). Its role includes promoting health, preventing diseases, and improving healthcare systems across its member countries in the Americas. PAHO consists of 35 member countries, including all countries in the Americas (North, Central, and South America, as well as the Caribbean). Each member country has an independent healthcare system and regulatory authority responsible for overseeing the approval and regulation of medicines within their respective territories. PAHO’s role primarily focuses on providing technical cooperation, guidance, and support to member countries in public health. It works closely with national health authorities, sharing information, expertise, and best practices to strengthen healthcare systems and address public health challenges. While PAHO does not have a centralized approval system for medicines like the European Medicines Agency (EMA), it provides guidance and recommendations on various health topics, including regulating and using medicines. [https://www.paho.org/ (accessed on 15 July 2023)].

The Pharmaceutical Inspection Co-operation Scheme (PIC/S) is an international organization that harmonizes Good Manufacturing Practice (GMP) standards and inspects pharmaceutical manufacturing facilities. Its primary objective is to ensure medicinal products’ quality, safety, and efficacy. The PIC/S operates through a collaborative framework involving regulatory authorities from multiple countries. The participating regulatory authorities exchange information, share expertise, and work together to establish common standards and guidelines for GMP inspection. The organization aims to promote consistency and quality in pharmaceutical manufacturing practices across borders. The PIC/S has 54 participating regulatory authorities from countries worldwide, including various European countries, Australia, Canada, Switzerland, Singapore, and others. Each member country is responsible for implementing and enforcing the GMP standards within its jurisdiction based on the guidelines and recommendations established by PIC/S. [https://www.picscheme.org/ (accessed on 15 July 2023)].

The International Conference on Harmonisation of Technical Requirements for Registration of Pharmaceuticals for Human Use (ICH) is a global organization that brings together regulatory authorities and the pharmaceutical industry to develop and promote harmonized guidelines for the registration, quality, safety, efficacy, and multidisciplinary evaluation of pharmaceutical products. The ICH aims to advance regulatory harmonization, streamline drug development, and registration processes, and facilitate global access to high-quality, safe, and efficacious medicines. The guidelines developed by the ICH provide a framework for consistent and efficient regulatory practices, allowing for more effective collaboration and information sharing among member countries and stakeholders in the pharmaceutical industry. The guidelines apply to the territories of the countries part of the ICH. These countries include the United States, European Union member states, Japan, Canada, Switzerland, and Australia. It is important to note that individual member countries may adopt and implement the ICH guidelines following their own regulatory frameworks and legal requirements. [https://www.ich.org/home.html (accessed on 15 July 2023)].

The World Health Organization (WHO) is a specialized agency of the United Nations (UN), the leading global authority on international public health. It operates to promote health, prevent diseases, and address health-related challenges on a global scale. The decisions and recommendations made by the WHO apply to all 194 member countries. Each member country is expected to align its national health policies and practices with the guidelines and recommendations put forth by the WHO. However, it is important to note that the WHO’s decisions and recommendations are not legally binding. Member countries can implement and adapt the WHO’s guidelines according to national contexts, legal frameworks, and health priorities. Nonetheless, the WHO’s guidance carries significant influence and is considered authoritative in global health, often shaping national health policies and practices worldwide. https://www.who.org (accessed on 15 July 2023).

The Gulf Cooperation Council (GCC) is a regional political and economic organization comprising six Arab states in the Arabian Gulf region. The GCC member countries are Bahrain, Kuwait, Oman, Qatar, Saudi Arabia, and the United Arab Emirates (UAE). The organization collaborates on various fronts, including healthcare, and has established a drug registration and regulatory framework. The GCC Central Committee for Drug Registration (GCC-DR) is the body responsible for overseeing the drug registration processes among member countries. The GCC-DR facilitates cooperation and harmonization in evaluating, registering, and post-marketing surveillance of pharmaceutical products across the GCC region. This process involves the formation of expert committees comprising representatives from member countries, who collectively review and evaluate the safety, quality, and efficacy data submitted by pharmaceutical companies. Once a pharmaceutical product is approved by one member country, it can be recognized and accepted by other member countries, thereby facilitating market access across the region. The GCC-DR has established guidelines, standards, and technical requirements for drug registration and regulation in member countries. Pharmaceutical companies seeking to register their products in the GCC region must comply with these guidelines and meet the specified requirements. https://www.gcc-sg.org/en-us/Pages/default.aspx (accessed on 15 July 2023).

The Association of Southeast Asian Nations (ASEAN) promotes economic, political, and social cooperation among its member countries in Southeast Asia. In drug regulation, the ASEAN has implemented a framework known as the ASEAN Common Technical Dossier (ACTD) and the ASEAN Common Technical Requirements (ACTR) for registering and regulating pharmaceutical products. The ASEAN member countries include Brunei Darussalam, Cambodia, Indonesia, Laos, Malaysia, Myanmar, the Philippines, Singapore, Thailand, and Vietnam. The ASEAN member countries have adopted a decentralized approach to drug registration. Each member country has a national regulatory authority responsible for evaluating and approving pharmaceutical products within its jurisdiction. However, the ACTD and ACTR provide a common framework that facilitates the mutual recognition of product approvals among member countries. Once a product is approved by one member country, it can be recognized and accepted by other member countries, enabling easier access to markets across the region; however, the member countries retain the autonomy to implement and enforce regulations according to their specific national requirements and legal frameworks. https://asean.org/ (accessed on 15 July 2023).

### 2.2. Functions of the GMA

The idea of a joint regulatory agency assisting multiple countries in expanding the availability of drugs is not new. Yet, none of these agencies listed above perform the functions anticipated by the proposed GMA charter that overcomes all constraints from a structured plan:Legally binding registration across member countries, regardless of their geography;No dossier sharing with member countries;Allows local labeling requirement compliance;Does not engage in price negotiations;The product must be registered in the country of origin;Automatic registration of SRA-approved products if they are also distributed in the country of origin;Rapporteur-based evaluation of registration dossiers from non-SRA countries;Third-party cGMP audit and testing sample surveillance;Centrally operated international scientific, legal, and technical expertise.

However, to make this proposal a success, a sponsor must set a role-model example that is difficult to find because states do not always agree with each other due to the differences in their perspective, their misunderstanding, and the desire for political independence. It is well exemplified by existing agencies that could meet the proposed GMA charter, but would not. However, if a model agency is established with an allowance for open membership, it can grow into the world’s most significant regulatory agency that focuses on biotechnological products.

## 3. Finding a Sponsor

The League of Arab States is proposed as a possible model for the GMA for multiple reasons based on the understanding of the medical anthropological landscape of these 22 countries, with a lot in common, from their historical evolution to today’s wealth disparity. To bring more relevance to the proposal, it is important to review the history of the members of the League of Arab States and their current constraints, whose realization will help bring about cooperation among the 22 Arab states.

### 3.1. History

Only a century ago, the colonial and military systems that ruled the Arab world’s territory at the time caused difficulties for the region’s healthcare sector [13,14]. Before the Second World War, most hospitals in the area were modest, private institutions founded by physicians who had completed their medical education overseas before returning to practice in their native nations. The legacy of colonialism [15] endured, even though many Arab republics had at least achieved a rudimentary degree of independence by the 1950s. This was especially true of the healthcare sector, where the paternalistic, top-down approach established during colonial administration persisted. The poorest citizens frequently received subpar government services. At the same time, those with means would often fly abroad for medical treatments or pay for care in private facilities—trends still prevalent today.

Since the countries in the region acquired independence some years ago, there has been much unrest in the area. Few Arab countries have thus far experienced the stability, openness, and affluence needed to establish an effective healthcare system. However, in the latter half of the 20th century, health indicators such as overall life expectancy and infant and maternal mortality rates unquestionably increased [16], largely because of a decline in local poverty, advancements in water, sanitation, and electricity systems, and a decline in the mortality burden of infectious diseases. Health outcomes [17] in the Arab world today, however, vary greatly due to economic, political, and social circumstances, with the best results being seen in affluent nations like Bahrain and Oman and the worst results being seen in unstable, low-income nations like Yemen and Somalia.

### 3.2. Affordability

The healthcare expenditure within the Arab League states varies from the lowest percentage of GDP of 1.8% by Djibouti to 8% by Lebanon [18]. The world’s highest GDP share for healthcare is the US at 18% [19]. (Figure 1) Of significance is the total per capita expenditure on healthcare, of which the portion for medicines is less than USD 20 per capita in Somalia, compared to USD 1400 in the US [20].

The inequity in the healthcare sector in the Arab World is well recognized [18]. But despite much effort by the Arab League, this diversity has been accepted as the fact that the health of most citizens across the region shall remain receiving less than optimal care [19], especially among the most marginalized countries, and little has been done to address these issues.

While most analysts will conclude from the correlation shown in Figure 1 that countries that have more wealth spend more on healthcare, there is also a possibility that countries that spend more on healthcare are also wealthier, a conclusion that still awaits proof.

### 3.3. Markets

The pharmaceutical markets in Arab states have rapidly grown in recent years [24], and by 2025, it is predicted to grow from its current level of USD 36 billion to approximately USD 60 billion [25]. Still, this represents a small and disproportionate portion of the worldwide pharmaceutical sector [26].

Saudi Arabia, whose pharmaceutical industry had a market value of more than 10 billion USD in 2021, anticipates a growth of about 7% during the following few years. Still, only a small part of this market will be filled by the pharmaceutical industry in Saudi Arabia [27]. However, the industry growth is imminent based on the plans based on Saudi Arabia’s Vision 2030 [28] effort, which stresses localization across various industries. Saudi Arabia also has a well-developed regulatory authority that could easily acquire the additional role of the GMA.

Another nation with a rapidly expanding pharmaceutical industry is Tunisia, which opened one of the region’s first pharmaceutical warehouses in the late 1930s [29]. This industry grew by more than 45% between 2014 and 2018, while exports rose by 7% during the same time. One hundred and twenty businesses, thirty-three actively creating drugs for human use, contributed to this growth. The nation has also significantly invested in training medical and pharmaceutical students, fostering collaboration with other pharmaceutical industries, and concluding direct supplier contracts.

Jordan is another significant player in the pharmaceutical industry and economic contributor. Jordan has a long history in the sector and is one of the few nations in the region with substantial pharmaceutical exports, primarily of generic drugs. In addition to meeting up to 25% of its own population’s medical needs, which makes the country less dependent on imports than most of its neighbors in the region, Jordan has a long history in the sector. Pharmaceutical exports in Jordan surpassed 1 billion Jordanian dinars (about USD 1.4 billion) in 2021, making them the only industry in the country to sell more than it imports. Approximately 75% of the pharmaceutical items made in Jordan today are exported [30]. This exceptional perspective in Jordan is ideal for fostering the production of biological pharmaceuticals, which brings far better profit margins and significantly boosts Jordan’s economy.

In recent years, the Arab countries’ market has seen a noticeable increase in the value share of biologics, in line with general industry trends, rising at a 30 percent annual pace, reaching close to 10 billion USD. With over USD 2 billion in sales, the Kingdom of Saudi Arabia (KSA) dominates the market. The following three countries are Algeria, Egypt, and the United Arab Emirates (UAE), with about 450 million USD in sales [31]. Healthcare costs, the GDP, and the demand for affordable therapies trigger this anticipated market expansion [31,32].

Article II of the Charter of the Arab League [33] identifies “health affairs” as its main priority; Article IV describes a mechanism for how the goals of the League are managed. However, there is a need to develop a more formal platform, as proposed in this paper, and it will be possible for the League to consider this suggestion to ensure better accessibility of lifesaving medicines [34].

To improve Arab integration in medicine control and law and to encourage knowledge sharing and best practices among Arab medicine control authorities, the Saudi FDA recently convened the first meeting of Arab regulatory authorities [35]. It was stressed that easing the registration and availability of pharmaceutical products, reducing their cost, and encouraging patient access to these treatments require cooperation and alignment with the legal and regulatory requirements for medicines. To track the application of Resolution No. 17, which was adopted at the 58th session of the Council of Arab Ministers of Health in March 2023, the Technical Committee for Arab Medicines also convened a concurrent meeting.

Saudi Arabia accounts for the largest market share among Arab countries. It is also a member of the Gulf Cooperation Council (GCC) that follows its centralized procedures in which different authorities are involved. The Gulf Health Council (GHC) acquires pharmaceutical products with proven efficacy, quality, and safety. The Gulf Central Committee for Drug Registration (GCC-DR) oversees various procedures, from manufacturing site registration to post-marketing surveillance.

The Saudi Food and Drug Authority (SFDA) reviews and approves biological products and their prices before they enter the market. The SFDA regulatory framework follows the United States (U.S.) FDA and the European Medicines Agency (EMA) guidelines with specificities that accommodate the local and regional (GCC) requirements.

The UAE is a member of the GCC and has similar requirements for biosimilar registration, including information on manufacturing consistency, immunogenicity demonstration, heterogeneity assessment, safety and efficacy studies, therapeutic equivalence, and a pharmacovigilance plan in which the public can voice their concerns directly to the Ministry of Health. Products produced in the UAE for international markets follow international guidelines set by the EMA and the World Health Organization (WHO). However, for local markets, they follow UAE standards and guidance developed by the GCC.

While there are some similarities regarding the cooperation among countries in the GCC, the scope of the GMA is very different; it is a global plan with specific targets and methodologies to achieve them.

## 4. Regulatory Misconceptions of Arab Countries

While the GCC brings the GMA model practice, albeit only partially, the first step needed to convince the Arab countries is to remove their misconceptions about the regulatory process, as identified below.

The 2nd Middle East North Africa (MENA) Stakeholder Meeting on Regulatory Approval, Clinical Settings, Interchangeability, and Pharmacovigilance of Biosimilars was conducted in Dubai, United Arab Emirates, on 10 October 2018, following the first MENA stakeholder meeting on biosimilars, which took place in Dubai in 2015. However, the remarks made at this meeting reveal misunderstandings that require correction, as presented below in italics [36]: “Regulatory bodies need scientists to evaluate a dossier. Others suggested that the region has plenty of scientists, pharmacists, and academicians to serve this duty”.*There is a great misunderstanding among Arab agencies about the qualification of regulatory staff. These include analytical chemists, pharmacokinetics, clinicians, statisticians, quality assurance auditors, lawyers, and others with specific expertise. Unless trained in a particular function, pharmacists, academicians, scientists, and politically appointed heads of the agency should not be part of any dossier evaluation. For this reason, I recommend installing a system of rapporteur evaluation of all biosimilar submissions. In addition, the FDA employs 11,000 full-time scientists* [37]*, so it is not expected of any non-SRI agency to be able to evaluate a regulatory dossier on its own.*“Discussion between national and international regulatory bodies is needed to ensure biosimilar approval is consistent worldwide”.*This is a broader goal that has never been possible, even for generic chemical drugs; there are agencies like the ICH and WHO that provide guidelines that are often adopted, but to expect that regulatory bodies will agree on issues that take their authority away, is not likely to be happy. However, this is what I am suggesting, but with a narrow goal.*“The limitations faced by recently established regulatory bodies must be recognized and addressed by mature regulatory bodies worldwide”.*What is meant by a “mature” authority; an “immature” authority should not operate in the first place. Expecting the FDA or EMA support can only be limited to following their guidelines. Suppose it is meant that the region’s authorities have been operating longer to help the newcomers. In that case, this, too, is misleading, for a more extended operation does not necessarily mean maturity.*“Countries with greater experience must support countries with less experience of biosimilars”.*It is doubtful that any country in the region has the required experience to make them a teacher. Therefore, the right thing to do is to harmonize the registration process, where a single agency approves biosimilars that all member countries will accept.*“Action should be taken to ensure that all biosimilar products globally are traceable at batch level to ensure adequate pharmacovigilance is upheld. Biosimilar naming will be key to this”.*This is not an issue; all products have registration and batch numbers, and the brand naming system is widely accepted to ensure traceability.*Strong governmental regulators should be in place to ensure drug products can be tracked.*Traceability is a fundamental process that applies to all drugs, which is the essential function of any regulatory authority.*“The long-term effects of switching and multi-switching between biosimilars and/or reference products need to be understood and addressed. This requires a concerted international effort to develop an optimal methodological approach”.*This is a misconception; there is no risk in interchangeability that is only an issue in the US, and that, too, is about to be removed. Therefore, there is no need to dwell on this wasteful exercise* [38].“Biosimilar patient registries could be established and implemented to gather further data on switching”.*It is not necessary. The EMA has recently reasserted this position allowing switching with the reference product and other biosimilars* [39]*. Most other countries in the rest of the world have already begun this practice that remains in the US due to legislative matters* [40].“Electronic healthcare records need to be developed and implemented to facilitate pharmacovigilance and gather further data on switching”.*Member countries can’t have electronic health records; pharmacovigilance is a common practice for all drugs. So, there is no need to extend this to switching.*“Encourage meeting with clinicians to explain what biosimilars are and what they are not, to enable them how to decide whether to prescribe biosimilars”.*Clinicians are least qualified to understand the nuances of the regulatory process; they must trust agencies’ decisions. No need to waste time teaching clinicians.*“Physicians, pharmacists, regulators, patients, and all stakeholders must communicate and share their experiences—challenges, and successes—with biosimilars”.*This wasteful exercise is more of a slogan; there is no need to teach or promote biosimilars. All must accept an approved biosimilar; promoting its safety and efficacy may even cause doubt about its safety and efficacy.*

Other misconceptions are summarized as follows:
Interestingly, except for Iran, which is not part of the Arab countries, all other countries in the Arab countries require clinical efficacy testing [32]. In addition, all Arab World regulatory authorities follow the FDA or EMA, and Egypt also includes the WHO [41].*The WHO is not a regulatory agency; it is an Agency whose decisions are widely criticized since it operates mostly on common consensus. The FDA and EMA guidelines depend on many legislative issues and legal exposure and are slow to change the guidelines as new science teaches otherwise.*A lack of agreement exists among the Arab countries regarding regulatory approval issues, particularly regarding interchangeability and switching. In Saudi Arabia, biosimilars are not automatically interchangeable. For example, ten biosimilars have been approved, of which only two are regarded as interchangeable. It is evident that in Saudi Arabia, biosimilarity alone is not sufficient for substitution or switching. However, biosimilars approved by the EMA are considered interchangeable. Additionally, a clinical trial that involves switching must be run to approve switching, which must happen before the biosimilar is approved.*None of these considerations are needed; as allowed in the EU, all biosimilars are interchangeable, even one biosimilar with another. The issue of interchangeability is indigenous to the US and is up for removal.*The Egyptian Drug Authority (EDA) “Guideline for registration of Biosimilar products in Egypt” is in place as of March 2020. The applicant must exhibit and compare the biosimilarity of their product to the innovator/reference product by completing and comparing pre-clinical and clinical studies and quality exercises. The EDA adopts the EMA guidelines and refers to the U.S. FDA’s safety and quality considerations, the WHO guidelines for evaluating similar biotherapeutic products, and relevant ICH (The International Council for Harmonisation of Technical Requirements for Pharmaceuticals for Human Use; https://ich.org/ (accessed on 14 July 2023) guidelines. In Egypt, however, the Ministry of Health will make interchangeability decisions, where the patient will not be given a choice.*Misconceptions regarding blanket following lead to archaic animal toxicology testing and other quality assessments that may not be necessary, as described below. An agency should create its guideline, albeit borrowing from any additional guideline, instead of listing another as the marker. Comments for interchangeability apply as stated above.*The Jordanian FDA’s guidelines are based on the EMA, where the EMA model has been implemented for quality assessment and comparability. It also authorizes the approval of manufacturing sites as a prerequisite to product approval and filing. Currently, six products have been approved according to Jordan’s biosimilar guidelines. Jordan’s approach to biosimilar regulation can be considered vigilant and strict. Nonetheless, biosimilars manufactured and marketed in reference countries, including but not limited to the UK, USA, Germany, France, the Netherlands, Sweden, Australia, Austria, and Japan, are usually given more privileges.*The same comments as offered for the Egyptian guideline apply here. In addition, references to biosimilars from some SRA countries should be expanded and automatically allowed registration without reviewing the dossier.*Medicines in Tunisia are obtained by centralized pharmacy purchase (PCP). The Biosimilar Specialized Committee makes decisions on a case-by-case basis regarding interchangeability. The committee comprises representatives of pharmaceutical inspection, a national control laboratory, regulatory authorities, various clinicians, and experts who utilize biosimilars.*The interchangeability issue is redundant; the approval committee need not include anyone who is not qualified to judge the compliance of a dossier. Moreover, approval should not be based on consensus, a significant weakness in almost all Arab state agencies, as elaborated above.*

## 5. Biosimilars

Therapeutic proteins are the only category of biological drugs that can be copied as biosimilars. A major focus of this paper was to enable faster entry of therapeutic proteins, either as new products or as biosimilars. While new products should be allowed without a detailed review of the dossier if they are marketed in an SRA country, biosimilars from non-SRA sources need an outsourced evaluation process, as detailed later in this paper.

While the list of biotechnological products that are direly needed is long, one class of products, recombinant biological drugs, makes an excellent choice to test the proposed regulatory plan. These drugs represent the fastest growing contributor to pharmaceutical sales, and their high prices also offer opportunities for Arab state manufacturers to bring biosimilars to these products that can be highly profitable (Table 2).

All these drugs can be made available as biosimilars, reducing their price by more than 80% while keeping a 90% profit margin. These calculations are based on the WHO findings that all monoclonal antibodies can be manufactured for USD 95–200/g [24]. Furthermore, these costs will decrease as newer technologies, such as continuous manufacturing [25], are adopted. This perspective should greatly interest companies within the Arab countries to consider domestic manufacturing and exports to compete in multibillion USD markets.

The market for biosimilars in the Arab World (Middle East and North Africa) is experiencing significant growth. According to a report by IQVIA, the biosimilar market in the Arab countries is projected to reach USD 2.17 billion by 2026, with a compound annual growth rate (CAGR) of 31.6% [26].

Recombinantly created biological agents, such as bacteria, mammal cells, and the like, are used to manufacture therapeutic proteins, which is why they are referred to as biological pharmaceuticals. After comprehensive safety and effectiveness testing, a novel biological drug is licensed and described, but not compared to existing medications. Contrarily, a biosimilar is accepted based on its similarity to the reference product; if the chemical and biological drug structures were identical, biosimilars would be accepted as chemical generics, and no analytical assessment would be necessary. The fundamental understanding of biologics lies in their 3D structure, which is responsible for their receptor binding and immunogenicity. So, logically, if it can be proven that the 3D structure of a biosimilar is exact (almost identical), it should reduce the testing significantly [42]. However, the expressed protein is subject to post-translational modifications in all expression systems, though more intensely in mammalian cells [43].

In 2006, the EMA approved the first biosimilar guidance and approved the first biosimilar [44]. As of April 2023, 47 biosimilars were approved in the US, including peptides, and 74 in the EU, representing 19 molecules, including peptides, out of more than 260 available recombinant therapeutic protein molecules [45] available as possible choices for biosimilars.

The EMA and FDA have modified the biosimilar approval guidelines over time as more evidence about their safety and efficacy has become available. The WHO also publishes guidelines to assist its 194 country members [46], but the WHO is not a regulatory agency; many member countries create their guidelines by “cherry picking” the WHO advice [47,48], risking the safety and efficacy of their biosimilars. For example, the Indian guidelines based on the WHO guidance [49] continue to include extensive animal toxicology testing and require efficacy testing in the local population on a fixed number of redundant and irrelevant patients. The proposal of the GMA is not redundant to the role of the WHO. The GMA is a regulatory agency with volunteer members who will accept the approval by the GMA unequivocally. The same comments apply to the ICH, a scientific body, not a regulatory agency, that does not approve products.

The first tranche of biosimilar approval guidelines treated biosimilars like new biological drugs with an abundance of caution, including extensive analytical comparisons, animal pharmacology and toxicology, clinical pharmacology, and clinical safety and efficacy studies. The only concession allowed was the extrapolation of indications. A comparative clinical efficacy test in one indication would be sufficient to qualify for all indications allowed for the reference product. To further assure safety and efficacy, biosimilars must have the same dose, strength, route of administration, and mechanism of action; the formulations may differ. Also, the prescribing information must be the same, and guidelines are available on writing the prescribing information for biosimilars [50].

Over time, the agencies became more convinced of the safety of biosimilars in response to challenges made to the guidelines [51]. It became well accepted that animal testing of biosimilars is redundant [52] given that even new biological products may not be required to conduct such testing because the mechanism of action of biological drugs involves receptor binding that is often unavailable in animal species [53]. The value of clinical efficacy testing has also come under criticism for scientific reasons since these studies cannot fail [53] and, if used to overcome a lack of similarity in analytical or clinical pharmacology, create a higher safety risk possibility if these studies are considered for approval.

An excellent example of progressive changes to guidelines comes from the MHRA. Last year, as the Brexit transition period ended, the MHRA (the Medicines and Healthcare Products Regulatory Agency) [54] published its first comprehensive guideline on 14 May 2022 that breaks from all other guidelines by providing clear judgment for not requiring animal and clinical efficacy studies [55].

Clinical pharmacology studies, including pharmacokinetic and pharmacodynamic comparisons, are a part of analytical methodologies, where similarities are established based on how the body is impacted by the drug and vice versa. These should be enhanced and recommended for newer technologies and approaches to develop structural equivalence.

Several ICH guidelines provide scientific support for developing biosimilars, which should be part of every guideline [39]. There is a dire need to harmonize the regulatory guidelines [56], but it is not likely to happen, as evidenced by historical events; for example, the guidelines for approving generic chemical drugs remain diversified for more than fifty years since chemical generics were introduced [57]. Moreover, countries do not agree on which oral product should have a waiver of bioequivalence study; Japan denies all. [58] So, it is understandable why harmonization and global concurrence can be challenging for a class or products as complex as biologics. It has little to do with science, but the legislative nature of these guidelines and the perspective held by the agencies are often difficult to change.

Since biosimilars have been around for 18 years, with hundreds of published reports on their safety and efficacy, a strong opinion has emerged [58] that significant amendments to the approval guidelines for biosimilars must be made, not only to reduce the development cost but also to enhance the safety of these products. Furthermore, lowering the development cost is essential to bring more biosimilars, as only nine out of more than one hundred and fifty possible biosimilar molecule candidates are approved in the US and fourteen in the EU. In addition, there are over 200 molecules that could provide excellent accessibility to patients [44]. Despite many efforts, there remains a dire need for consolidated regulatory guidelines [39]. Any consolidation should be based on the current scientific understanding to remove unnecessary and irrelevant testing, as the FDA acknowledges [57,59]. These steps are essential to reduce the current development cost of biosimilars, which range from USD 100 to 300 million [60].

Other misconceptions include animal testing [51] and clinical efficacy testing in patients [53]. At the end of 2022, the US government passed a new law, The FDA Modernization Act 2.0 [61], removing the term “animal toxicology” and replacing it with “nonclinical” to remove all animal testing since animals do not have the receptors to respond to biological drugs. In addition, the MHRA recently announced that animal and clinical efficacy testing might be unnecessary [54]. This will be the first requirement for any universal guideline to remove animal testing; if used to justify the variability in analytical assessment, as is commonly practiced, animal testing creates a risk of approval of unsafe biosimilars.

For biosimilars, higher immunogenicity compared to that of the reference product can be a major issue, primarily if anti-drug antibodies (ADAs) are formed and if the ADAs alter the disposition profile of the drug. For example, in the case of highly immunogenic insulin, the FDA does not require immunogenicity testing since the disposition profile is not altered despite differences in immunogenic responses. In vitro, immunogenicity assays might even be recommended as part of the functional, analytical assessment, though they do not replace the immunogenicity assessment in the PK study. Another understanding that short-term immunogenicity analyses might not correspond to real-world use where rare ADA-related events might become evident remains under consideration.

The limitations of efficacy testing in patients are well recognized by regulatory agencies. To overcome these concerns, the FDA’s Division of Applied Regulatory Science (DARS) [62,63] has recently published its recommendations to remove efficacy testing in patients for biosimilars [64] based on a comparison of pharmacodynamic (PD) markers, labeling it as “clinical efficacy testing in healthy subjects”. A PD biomarker is not required to be a surrogate endpoint or to have an established relationship with clinical efficacy outcomes [63,65]. Additionally, clinical efficacy testing in patients can result in the approval of unsafe products if used to overcome analytical and clinical pharmacology profiles mismatches, as they are substantially more sensitive and objective tests; for example, the clinical efficacy objective of the duration of severe neutropenia is less sensitive than the PD biomarker, the area under the effect-time curve of an absolute neutrophil count [52].

DARS made these conclusions based on its investigations [66] and clinical studies conducted for this specific purpose [67,68] to define the best practices for characterizing the PD biomarkers for various drug classes. These studies evaluated the use of human plasma proteomic and transcriptomic analyses to find novel biomarkers for the approval of biosimilars [69]. More efforts are underway to remove patient testing for all biological drugs, including monoclonal antibodies that do not show pharmacodynamic markers [70]. The gold standard for evaluating the clinical efficacy of novel medications compared to placebos has come under fire recently. Dr. Janet Woodcock, a past acting commissioner of the FDA, has stated: ‘Why should we put patients through all these different trials just to check a box.’ The FDA has recently questioned this idea of real-time testing, claiming that clinical efficacy testing is “broken” [71]. Following the 21st Century Cures Act, new digital technologies, and real-world evidence (RWE) are necessary [72]. Recently, the FDA has announced policies and funding to encourage the development of novel clinical trials and substitute trials with non-clinical methodologies [73].

While the role of efficacy testing in patients shall remain controversial, such testing for biosimilars is questioned on many grounds. For example, so far, all such trials of biosimilars have reported no clinically significant differences, leading to the approval of all products that reached this development stage, as shown in the 96 EPAR files from EMA [74] and 37 approval documents from the FDA [75]. In addition, the research published on the clinicaltrials.gov website [76] substantiates that all 141 studies met the acceptance criteria. The PubMed database also provides the results of 435 randomized control clinical trials conducted between 2002 and 2022 that showed no clinically significant difference [77].

The standards for surrogate biomarkers used to support the approval of novel drugs are fundamentally different from the standards for PD biomarkers meant to assist in the demonstration of biosimilarity [78]. This provides opportunities for biomarkers to be used as secondary and exploratory endpoints in new drug development programs to support biosimilar testing. In addition, many opportunities are available to identify new PD biomarkers or fill information gaps on existing biomarkers to facilitate the use of PD biomarker data in clinical pharmacology studies instead of comparative clinical efficacy studies.

Examples of drugs that exhibit pharmacodynamic markers and are thus exempt from patient testing if other attribute comparisons are found acceptable are presented in Table 3.

For products that do not display PD biomarkers, such as monoclonal antibodies, other “omic” technologies like transcriptomics and metabolomics may offer a chance to find new, sensitive, and robust candidate biomarkers for further exploration as PD biomarkers [79]. However, a more rational approach will be to take a step back in the testing cycle of biosimilars and examine if ex vivo testing can provide evidence of biosimilarity that is more sensitive and reliable in identifying any “clinically meaningful difference” in the language of the FDA guidelines.

Since the pharmacodynamic response is triggered by receptor binding, cell-based bioassays or potency assays, such as ELISA, binding assays, competitive assays, cell signaling, ligand binding, proliferation, and proliferation suppression, should provide a good functional comparison of a biosimilar candidate with its reference product. Furthermore, functional tests for the mode of action (MOA), such as testing for apoptosis, complement-dependent cytotoxicity, antibody-dependent cellular phagocytosis, and antibody-dependent cellular cytotoxicity, are generally not required and can be added to provide a higher degree of confidence in safety and efficacy.

Monoclonal antibodies (mAbs) bind to specific protein epitope targets, resulting in a therapeutic response. Characterizing the mAb’s affinity for binding includes target antigen and affinity for binding to specific Fc receptors (Fc (RI, Ia, IIa, IIb, IIIa, IIIb; Fc(RN))), Effector functions like ADCC and CDC, molecular properties like charge, pI, hydrophobicity, and glycosylation, and off-target binding employing in silico or in vitro techniques like baculovirus ELISA tools are all robust and objective to establish functional similarity [80,81]. Additional tests can be added based on specific applications such as for TNFα blockers: C1q; CDC; induction of regulatory macrophage; inhibition of T-Cell proliferation (MLR); LTα; MLR; mTNFα; Off-target cytokines; Reverse signaling; sTNFα; Suppression of cytokine secretion; and tmTNF-α. The functional assays form more robust markers to establish efficacy comparisons than testing in patients, without the necessity to demonstrate any PD response for mABs [82,83]. However, the functional tests (ADCC, ADCP, and CDC) are of little value when the drug targets a soluble antigen [84,85].

A collection of functional assays pertinent to a range of biological activities can be employed for a product having multiple biological activities. For instance, some proteins have a variety of functional domains that express enzymatic and receptor-binding functions. The metric for biological activity is potency. Analytical studies to evaluate these features are easily accessible when immunochemical properties are made part of the activity assigned to the product (for instance, antibodies or antibody-based products). The functional assays form more robust markers to establish efficacy comparisons than the testing in patients, without the necessity to demonstrate any PD response for mABs [86,87,88].

In May 2023, the FDA issued draft guidance, “Generally Accepted Scientific Knowledge in Applications for Drug and Biological Products: Nonclinical Information” [89], suggesting that nonclinical testing can be reduced based on GASK: first, where a product contains a substance (either naturally derived or synthesized) that occurs naturally in the body and has known effects on biological processes; and second, where a sponsor has demonstrated a drug’s impact on a particular biological pathway to conclude that certain nonclinical studies are not necessary to support approval and labeling of the drug. For example, some drugs have distinct effects on well-known biological pathways; therefore, specific outcomes can be predicted once the drug’s effect is demonstrated on the biological pathway. In addition, in some cases, a drug has either on- or off-target impacts on a biological pathway or a molecular mechanism of action that is known to result in adverse effects at clinically relevant exposures based on the operation of the biological pathway. Thus, according to the FDA, it may be appropriate to rely on GASK regarding the impact of the pathway rather than to conduct specific pharmacology and/or toxicology studies intended to measure the impact of the path.

The recent FDA Modernization Act [61] that amends the Biological Products Competition and Innovation Act (BPCIA) [90] has removed the term “animal toxicology” and replaced it with “nonclinical” testing to assert that unnecessary testing of biological drugs that act by receptor binding, and thus do not display animal toxicology, is not necessary.

If testing in humans cannot result in any useful information, it becomes an ethical concern, as codified in the US 21 CFR 320.25(a)(13), forming the universal belief that “No unnecessary human testing should be performed” [91].

## 6. Proposed Guideline for Biosimilars

The discussions presented above form the basis of a rational regulatory guideline that is recommended here for the GMA to adopt. Since the guideline’s scope can change over time, it is more practical to revise a single guideline periodically.

Since the purpose of the GMA establishment is to enhance the entry of modern therapies into its member states, the approval process is divided into two classes of product definitions: products imported from Stringent Regulatory Authority (SRA) countries [92] and the other for products manufactured in a non-SRA country.

### 6.1. SRA Sourcing

The World Health Organization (WHO) [92] defines an SRA as applying stringent requirements for quality, safety, and efficacy in its regulatory examination of pharmaceuticals and vaccines for marketing authorization, an idea created by the Global Fund to Fight AIDS, Tuberculosis, and Malaria to help with decisions about purchasing pharmaceuticals for humanitarian aid. This concept allows drug authorities to expedite the registration or marketing authorization of medications that have already received SRA approval. As of 2022, the current list comprises 36 nations. The EC members include Austria, Belgium, Bulgaria, Croatia, Cyprus, Czech Republic, Denmark, Estonia, Finland, France, Germany, Greece, Hungary, Ireland, Italy, Latvia, Lithuania, Luxembourg, Malta, Netherlands, Poland, Portugal, Romania, Slovakia, Slovenia, Spain, and Sweden. Other nations include the United Kingdom, Iceland, Liechtenstein, Norway, Japan, the United States of America, Canada, Switzerland, and Australia.

The GMA approval process should be made more efficient by automatic registration with only summary data without including any proprietary information if the following conditions are met: the same label (indications and description) as approved in the country of origin, and the batches supplied should come from the same batch distributed in the country of origin.

The products imported from the SRA countries can also be biosimilars.

### 6.2. Non-SRA Country Biosimilars

Registration dossiers from non-SRA countries require extensive scrutiny for compliance, data, and business practice integrity [93,94,95,96].

The definitions of the terms are provided below:

Qualified product: a product with a reference SRA product currently distributed in the country of origin; the proposed biosimilars should have the same mode of action, dose, frequency, route, and concentration (strength).

GMA review: when a dossier is submitted, it is reviewed by regulatory experts to ensure that it is complete; at this stage, it is not a scientific review, only a compliance review intended to reduce the chances of rejection by a rapporteur.

Rapporteur: Using rapporteurs is a standard practice in the EU; the FDA also accepts third-party audits [97]. Rapporteurs are members of the Committee for Medicinal Products for Human Use (CHMP) or the Committee for Medicinal Products for Veterinary Use (CVMP), assigned to assess applications for marketing authorization. They play a critical role in evaluating and monitoring medicines in the EU. Competent national authorities of the EU Member States appoint the rapporteurs. The EMA generally identifies the rapporteurs and co-rapporteurs for specific medicines in its assessment reports, keeping the identities of the rapporteurs and co-rapporteurs confidential in certain situations. For example, a list of 61 rapporteurs for biosimilars is available at the EMA [98]. In addition, the GMA should create a list of global rapporteurs. The cost of the rapporteur is paid by the GMA and charged to clients to avoid any direct contact.

Third-Party cGMP (current Good Manufacturing Practice) Audit: data and sample integrity: Since the clinical pharmacology testing for biosimilars is conducted in an at-scale cGMP lot (meaning a final commercial lot), it is imperative that the developer qualifies its cGMP production. The audit is specific to the product and is not waived based on previous audits. The audit is conducted by third-party auditors, not by any GMA staff or other member agencies. The auditors also confirm and assure that the samples going out for clinical pharmacology testing are valid and their integrity is confirmed.

Validated Samples: The samples used for analytical assessment and clinical pharmacology must be validated for their source, history, and compliance. Generally, an audit will collect these samples and provide them to the third-party testing facility.

Third-Party Analytical Assessment: The final analytical assessment must be conducted by a third party approved by the GMA as a qualified testing facility.

Certified CRO Samples Retained Clinical Pharmacology: CROs should retain the samples if there is an issue regarding an outlier or later inquiry; the time limit is through the product’s shelf life.

Reference Product: To qualify as a reference product, a comprehensive dossier must have approved a biological product that is still being marketed in the nation of origin. There can be only one reference utilized. When different strengths or presentations of the reference product are available, the lowest-strength product should be used. Several batches of the reference product should be used having other times on the market and obtained directly from the market. The reference product batches should be tested for attributes to establish the shelf life and stored as recommended. It may occasionally be possible to test batches that have been stored for a long period (for example, frozen at −80 °C) or past their intended shelf life if reliable data demonstrate that the storage conditions have no impact on the critical quality attributes.

Characterization: As defined in ICH Q6B, proper methods characterize the reference product. Some of these characterizations determine the physicochemical qualities, biological activity, immunochemical properties (if any), purity, impurities, contaminants, and amount. Developers are encouraged to adopt newer technologies as available. Since the quality attribute values of the reference product can vary from batch to batch, it is essential to establish the ranges of these variations. The variations are either process-related (the manufacturing system) or product-related (the expression system); the latter often cannot be resolved, requiring the developer to create a different expression system; the same can be the case for process-related attributes, but these are readily fixed. However, any differences in both groups of attributes cannot be justified based on any in vivo or ex vivo studies [41].

Impurities: When developing a biosimilar, impurity profiling is required, and guidelines for product-related variations with the innovator are established. For instance, a biosimilar may show fewer impurities in terms of type and quantity. Still, there must be no mismatched impurity, as it cannot be justified by a safety study.

Function-based tests: Critical quality attributes (CQA) should be identified using analytical and in vitro functional levels. Functional experiments should be pertinent to the potential MOA in all therapeutic indications, including those that examine apoptosis, complement-dependent cytotoxicity, antibody-dependent cellular phagocytosis, and antibody-dependent cellular cytotoxicity. Functional tests (ADCC, ADCP, and CDC) are unsuitable for a reference product primarily targeting a soluble antigen.

Test procedures: Testing of CQAs does not require validated procedures, as some test methods cannot be fully validated. Analytical methods must be qualified, sensitive, and adequately selective to identify potential differences. Where appropriate, the procedures described in the ICH recommendations (ICH Q2A, Q2B, Q5C, and Q6B) for analytical assessment can also be utilized to evaluate the quality attributes for batch release. Additionally, the use of appropriate orthogonal methodologies is necessary for robust data.

Number of batches: Generally, eight batches will be tested; one should be a clinical batch. Therefore, the final third-party analytical assessment will include at least three PPQ lots.

Data Evaluation: Depending on the type of data output, a visual comparison suffices for test results sent as printed outputs, such as spectra. Quantifiable data from multiple batches should use the 3Sigma range derived for the reference sample as (ref − 3ref, ref + 3ref), which provides the most accurate inference. If the test sample’s min–max range falls within the 3-Sigma range, the sample is accepted.

Expression System: The expression system determines the product-related critical quality attributes (CQAs), which include primary structure, higher-order structures (HOS), glycosylation (only in eukaryotic hosts), product-related variations, and process-related variants. The expression system should be of the same class as the one used to express the reference product, even though SRA agencies allow the use of a different expression system. This recommendation comes from the realization that switching an expression inevitably leads to variable post-translational modifications that may be difficult to evaluate for safety and efficacy. The developers are also advised to select more steady expression systems; generally, high-yielding cell lines produce more variants. Cell lines should be qualified according to the ICH Q5D.

Analytical Profiles. Proteins undergo the addition of functional groups after the translation, called post-translational modifications, that should be comparable, not necessarily identical. In addition to PTMs, these profiles include aggregates, fragments, visible or subvisible particles, acidic and basic variants, and other product modifications such as the reduced, oxidized, glycated, and misfolded forms. These attributes can change over the product’s shelf life, requiring testing over the shelf-life duration. When the environment changes during different stages of the production process, the hydrophobic parts of the protein can unfurl, causing accumulation or fragmentation, adding to immunogenic responses. The aggregate size ranges from soluble aggregates to visible residues, depending on the duration of exposure to various stresses such as shear, thermal, chemical, freeze-thaw, etc. The matrix-free SEC substitute analysis helps define the size distribution that is further confirmed by sedimentation velocity-analytical ultracentrifugation (SV-AUC).

Charge variations are proteoforms that appear at different stages of the manufacturing process in various colloidal matrices (such as culture medium, in-process buffers, or formulations) and have varying charges. It is, therefore, preferable to use several types of cation exchange (CEX) chromatography.

Oxidation, phosphorylation, sulfation, acetylation, methylation, and hydroxylation are examples of non-enzymatic post-translational modifications (PTMs) created throughout various manufacturing stages. Liquid chromatography is preferable for defining PTMs and measuring the associated molecular variations and contaminants.

Cell substrates are process-related variations or residuals, including HCPs, HCDs, cell cultures, and downstream processing residuals. Enzyme-linked immunosorbent assays (ELISA) and real-time or quantitative PCR are the main HCP and HCD detection and quantitation techniques. These variants are not tested during the drug substance qualifying phase because they are part of the release specification.

Release Specification: Release specifications are based on the characterization of the reference product, except for the legacy compendial attributes such as sterility, fill volume, and delivered volume; other characteristics are independently established, such as sterility, invisible particles, protein content, potency, and physical characteristics unique to the biosimilar candidate. These standards may be used to specify the biosimilar candidate’s release specification.

Formulation: A formulation different from the reference product’s formulation is permissible for biosimilars. A formulation with the same number of inactive substances or fewer is advised unless constrained by patent protection. The formulation’s stability, compatibility (i.e., how it interacts with excipients, diluents, and packaging materials), and compatibility should all be proved, along with the active ingredient’s integrity, activity, and potency. If the primary packaging in touch with the product is different, further safety tests are required to verify that there is no unexpected leaching of package components into the product. Developers are encouraged to select a primary packaging material that is similar instead, as it is often difficult to defend these findings. The formulation may not contain any unique excipients previously not used in a similar product, and all excipients must be free of animal products.

Reference Standard: In-house primary reference material is an adequately documented sample made by the manufacturer from a representative lot or lots and calibrated against which in-house working reference material is used for biological assays and physicochemical testing of the following lots. It is the sole source acceptable for use as a working reference. Reference standards that are openly accessible (like European Pharmacopoeia) cannot be used as the reference product for comparison testing.

Stability: The stability of the biosimilar candidate must be evaluated per ICH Q5C, including accelerated and stress stability testing, to enable a direct assessment of structural similarities further and produce degradation profiles.

Process Qualification: Before any analytical assessment of similarity, the upstream and downstream processes must be checked. However, once the clinical pharmacology studies are finished, no batch size adjustment is permitted; the developer may only do this under ICHQ5E, which only applies post-approval. Bridging studies are needed to validate whether the production size changes.

Animal Toxicology: For biosimilars, no animal toxicological testing is necessary. This conclusion is based on the recent amendment to the BPCIA, removing “animal toxicology” and replacing it with “nonclinical testing”. Since animals have no receptors to bind with biological drugs, and this binding results in pharmacology and toxicology, this testing is now also recommended for new biological drugs.

Clinical Pharmacology: An extension of analytical evaluation, pharmacokinetic (PK), and pharmacodynamic studies (PD) reflect how the body perceives the drug molecule and vice versa. Such studies are also conducted for drugs like aflibercept or ranibizumab that are administered locally into the eyes; the drugs do not enter general circulation, so they are tested by administering parenterally for the same reason. For most chemical generic drugs, PK profiling is not required when administered intravenously, intramuscularly, or subcutaneously. However, biosimilars administered by parenteral routes require PK profiling since the pharmacokinetic parameters, such as half-life and distribution volume, can also correlate with the kinetics receptor binding, an essential assessment since all biological drugs act by receptor binding.

One misunderstanding in the design of PK/PD comes from the traditional goal to characterize the profiles in a wide range of subject qualifications such as age, gender, body mass index (BMI), body weight, and race. All these variables add much inter- and intra-subject variability that requires a larger population. None of these are necessary for comparative PK/PD profiling since these studies aim not to characterize but compare the profile attributes. A robust design should accommodate crossover or parallel designs. A crossover approach is better at identifying differences but might not be appropriate for reference products with robust immune responses or long half-lives. The equivalence margins must be pre-specified, and an appropriate range is often 80.00–125.00%. The key PK parameters, typically AUC0-Cmax, should be equivalent in the PK experiment.

Immunogenicity: Immunogenicity is an inherent property of proteins and is best tested in healthy subjects in clinical pharmacology profiling. However, it is important to note that the immunogenicity of a specific protein can be assessed through preclinical and clinical studies during drug development. These studies evaluate the protein’s potential to elicit an immune response, including the production of antibodies against the protein.

Clinical Efficacy: No clinical efficacy or safety testing is required for molecules with a pharmacodynamic response; this will exclude mAbs until similar waivers allow them. Comparative clinical efficacy testing requires hundreds of thousands of patients to be statistically meaningful; thus, such studies have never failed. Until the SRAs allow further waiver of clinical efficacy testing in patients, the GMA will waive all studies only for drugs with PD markers.

Naming: Biosimilars should have a brand name and share the same International Nonproprietary Name (INN) as the reference product and any additional designations required in the local jurisdiction. Biosimilars should also have different brand names.

Label: The label must, without exception, include all risks related to the reference product and have the same indications. The developer is not permitted to ask for fewer or more indicators.

Substitution: The reference product and other biosimilars authorized using the same reference product can be replaced or interchanged with biosimilars [98]. It was most recently confirmed by the European Medicines Agency (EMA).

Pediatrics: For biosimilars, no pediatric compliance studies are necessary.

Human Factors Studies: These investigations are necessary to ensure that the appropriate dose is given when a patient administers a product. However, these studies are not required if the device utilized is very similar to the reference product’s device. Furthermore, no such studies are necessary when a healthcare expert uses the product.

Risk Management: A biosimilar product uses the same risk management strategy (RMP) as the reference product. No pharmacovigilance is required for products supplied from SRA countries, where these products are also marketed.

## 7. Conclusions

Bringing newer technology biological drugs to developing countries is a challenge for multiple reasons. Despite many efforts by regional and global agencies, it remains a challenge to encourage the entry of newer drugs and to encourage manufacturing within non-SRA countries. The formation of GMA will resolve all constraints, giving developers a larger market and much-reduced submission requirements as an incentive, giving regional agencies technical support to approve the products manufactured locally, and creating a culture of technology adoption in developing countries. While several global agencies have promoted the same perspective, they have failed because of the lack of clarity in their charters, including the lack of mandatory acceptance of registration by member countries.

The League of the Arab States was chosen as a role model to create and expand the charter to any country ready to bind itself to the alliance. The GMA’s function is well defined in approving a little review for products from SRA countries and adopting a third-party evaluation of non-SRA dossiers to ensure transparency and remove doubts about these products’ safety and efficacy.

To bring granularity to the proposed charter, it is open to all products, including therapeutic proteins, gene therapy, CRISPR-Cas9, mRNA therapeutics, CART therapy, and many more. However, for non-SRA countries, it will be limited to biosimilars, since these are the only biological products that can be copied. Furthermore, this proposal is not to be compared with the role of other agencies with a global focus, since their charter does not require a binding commitment by the member states that expands the impact on of the GMA.

However, it will not be easy to secure the concurrence of the 22 state members of the League of Arab States without much political action by the leaders of these countries. It will also help if the recently formed agency, the African Vaccine Manufacturing Initiative (AVMI), which also focuses on biological drugs [99], decides to join hands with the League of the Arab States. This will expand the target population to almost two billion, making it more lucrative and bringing a faster adoption of the proposed charter.

## Figures and Tables

**Figure 1 healthcare-11-02075-f001:**
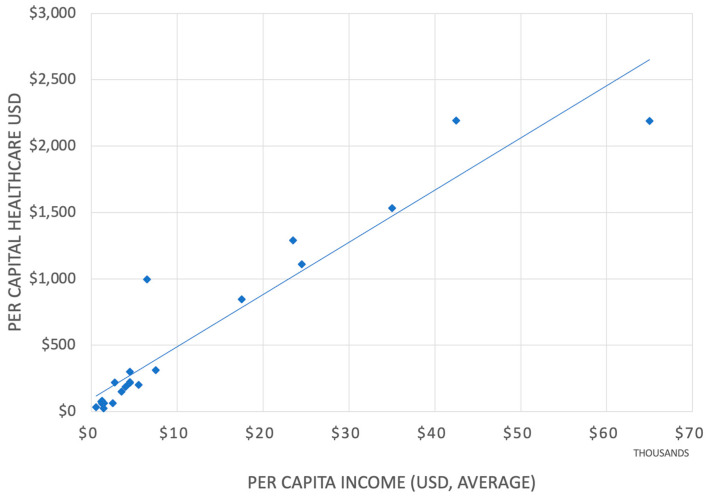
Per capita income and healthcare expenditure correlation within the 22 Arab states (Algeria, Bahrain, Chad, Comoros, Djibouti, Egypt, Iraq, Jordan, Kuwait, Lebanon, Libya, Mauritania, Morocco, Oman, Palestine, Qatar, Saudi Arabia, Somalia, Sudan, Syria, Tunisia, United Arab Emirates, and Yemen) with a population of over 400 million [21,22,23].

**Table 1 healthcare-11-02075-t001:** Top ten most expensive treatment costs till June 2023 [12].

Rank	Drug	Manufacturers	Indication	Cost
1	Hemgenix	CSL Behring, uniQure	Hemophilia B	$3.5 million/dose
2	Skysona	Bluebird bio	Cerebral adrenoleukodystrophy	$3 million/dose
3	Zynteglo	Bluebird bio	Transfusion-dependent thalassemia	$2.8 million/dose
4	Zolgensma	Novartis	Spinal muscular atrophy	$2.25 million/dose
5	Myalept	Chiesi Farmaceutici	Leptin deficiency	$1.26 million/year
6	Zokinvy	Eiger BioPharmaceuticals	Hutchinson-Gilford progeria syndrome and processing-deficient progeroid laminopathies	$1.07 million/year
7	Danyelza	Y-mAbs Therapeutics	Relapsed or refractory high-risk neuroblastoma	$1.01 million/year
8	Kimmtrak	Immunocore	Uveal melanoma	$0.97 million/year
9	Luxturna	Spark Therapeutics	Biallelic RPE65-mediated inherited retinal disease	$0.85 million/treatment
10	Folotyn	Acrotech Biopharma	Relapsed or refractory peripheral T-cell lymphoma	$0.84 million/year

**Table 2 healthcare-11-02075-t002:** Twenty top-selling drugs in 2022 [23]. Total sales USD 186.40 billion.

Drug Name	2022 Sales, USD Billion
Actemra/RoActemra (tocilizumab)	USD 2.58
Darzalex (daratumumab)	USD 7.98
Dupixent (dupilumab)	USD 17.42
Enbrel (etanercept)	USD 4.12
Eylea (aflibercept)	USD 12.72
Hemlibra (emicizumab)	USD 3.65
Humira (adalimumab)	USD 21.24
Imfinzi (durvalumab)	USD 2.78
Lantus (insulin glargine)	USD 2.38
Ocrevus (ocrelizumab)	USD 5.76
Opdivo (nivolumab)	USD 8.25
Perjeta (pertuzumab)	USD 3.90
Prolia (denosumab)	USD 3.63
Remicade (infliximab)	USD 2.34
Skyrizi (risankizumab)	USD 5.17
Stelara (ustekinumab)	USD 9.72
Taltz (ixekizumab)	USD 2.48
Tecentriq (atezolizumab)	USD 3.55
Tremfya (guselkumab)	USD 2.67
Trulicity (dulaglutide)	USD 7.44

**Table 3 healthcare-11-02075-t003:** Biosimilars with PD markers are exempted from clinical efficacy testing in patients.

Drug	Patent Expiry
Interferon beta-1b	2004
Parathyroid hormone	2004
Interferon alfa-2b	2004
Chorionic gonadotropin	2007
Interferon alfa-n3	2011
Etanercept	2012
Menotropins	2015
Urofollitropin	2015
Peginterferon alfa-2b	2015
Interferon beta-1a	2020
Insulin regular	2025
Insulin lispro	2014

## Data Availability

Not applicable.

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
