# Peer review of "A Proposed Global Medicines Agency (GMA) to Make Biological Drugs Accessible: Starting with the League of Arab States"

_healthcare, 2023, doi:10.3390/healthcare11142075_

Round 1

Reviewer 1 Report

The similarity index of the manuscript may be kindly be checked. 

The language may be improved and gramatical mistakes must be rectified. 

Author Response

Comments

Manuscript Number: healthcare-2461483 title “A Proposal to Establish a Global Medicines Agency (GMA) for Making Biotechnology Drugs Affordable Starting with the 3 League of Arab StatesHealthcare Journal

The author presented a reasonable proposal to establish a global model for biotechnology based drugs. The model will definitely produce cost-effective drugs. However there are some limitations and legal complications in the model which needs to be discussed in the manuscript with proper justifications. The idea of the model may be discussed with reference to the following concepts.

https://www.liebertpub.com/doi/10.1089%2Fomi.2017.0141 https://www.sciencedirect.com/science/article/pii/S0167779908001789 https://heinonline.org/HOL/LandingPage?handle=hein.journals/stanit45&div=12&id=&page= https://heinonline.org/HOL/LandingPage?handle=hein.journals/ambuslj45&div=30&id=&page= https://books.google.com.pk/books?hl=en&lr=&id=lZNxDwAAQBAJ&oi=fnd&pg=PP1&dq=to

+Establish+a+Global+Medicines+Agency+(GMA)+for+2+Making+Biotechnology+Drugs&ots= TeoS8aQAe7&sig=cB9OPpTSdtCPRU066yDjRMVIdws&redir_esc=y#v=onepage&q&f=false

I HAVE ADDED THE DISCUSSION OF MEDICAL ANTHROPOLOGY THAT WAS A GREAT IDEA; OTHERS WHERE BOTANICAL PRODUCTS ARE CONSIDERED I HAVE LEFT OUT.

In the abstract abbreviations must be avoided such as EMA. The full name with abbreviation must be used at first mention.

ALL SUCH ENTIRES ARE FIXED.

The author claims that animal toxicity and clinical efficacy studies might be removed then how the safety and efficacy profile will be established?

I HAVE MODIFIED IT TO ENSURE THAT THIS ISSUE IS NOT A CONSTRAINT; THE REASONS FOR REMOVING ANIMAL TOXICOLOGY IS WELL-ESTABLISHED AND NOW A LAW; THE EFFICACY TESTING ALWAYS FACES THE STATISTICAL CONSTRAINTS INCLUDING THE BAYESIAN THEORY BUT TO ENSURE THAT UNTIL THESE CONCEPTS RE WELL ACCEPTED, WE CAN WORK WITH WHATEVER IS THE REGULATORY STANDARD.

The bioequivalence studies of the biosimilars with reference product must be ensured to avoid the pharmacokinetics and pharmacodynamics issues and authenticate the qulity control of these biosimilars. How and who will this model be implemented? Similarly League of Arab States has genetic diversity due to huge number of people from across the world are living there, so some of the hypersensitivity and toxicity studies may be carried out to ensure the safety of these drugs.

The pharmacokinetic profiling is mandatory for these drugs in each and every ethnic group to know about the impact of genetic variation.

WHEN WE ARE CONDUCTING BE STUDIES, THESE ISSUES ARE RESOLVED BASED ON CROSS-OVER, AND THE SAME HOLD FOR EFFICACAY TESTING SINCE THE TWO PRODUCTS ARE TESTED SIDE BY SIDE, BOTH CLAIMING TO BE THE SAME.

The language must be improved. The style of the references needs to be uniform throughout. The typographical and grammatical mistakes must be rectified in the manuscript.

I HAVE FIXED THE REFERENCES AND ALSO HAD AN EDITING DONE ON THE MANUSCRIPT. THANKS FOR  POINTING IT OUT.

Reviewer 2 Report

see attached comments

Just watch for some incidents of first person and colloquial language. Specifics are noted in comments

Author Response

Please see the enclosed document that provided a line by line response to the reviewer's comments.
